# RoTipBot: Robotic Handling of Thin and Flexible Objects using Rotatable Tactile Sensors

Jiaqi Jiang*, Xuyang Zhang*, Daniel Fernandes Gomes, Thanh-Toan Do and Shan Luo

*Abstract*—This work introduces RoTipBot, a novel robotic system for handling thin, flexible objects. Different from previous works that are limited to singulating them using suction cups or soft grippers, RoTipBot can count multiple layers and then grasp them simultaneously in a single grasp closure. Specifically, we first develop a vision-based tactile sensor named RoTip that can rotate and sense contact information around its tip. Equipped with two RoTip sensors, RoTipBot rolls and feeds multiple layers of thin, flexible objects into the centre between its fingers, enabling effective grasping. Moreover, we design a tactile-based grasping strategy that uses RoTip's sensing ability to ensure both fingers maintain secure contact with the object while accurately counting the number of fed objects. The results show that RoTipBot not only achieves a higher success rate but also grasps and counts multiple layers simultaneously. The success of RoTipBot paves the way for future research in object manipulation using mobilised tactile sensors.

## I. INTRODUCTION

Thin and flexible objects pose two key challenges for robot manipulation: overlapping layers often obscure underlying states, leading to incomplete or noisy visual observations, and their deformability requires dexterity and compliance in robotic grippers. Existing approaches often bypass these challenges: some assume known object positions [1], while others use costly force/torque sensors to mitigate visual noise [2], increasing overall cost. To address deformability, methods like suction [3], [4] and two-finger soft grippers [2], [5], [6] are employed. However, these allow grasping only one page per closure, limiting efficiency. Suction cups struggle with air gaps, preventing vacuum seals, while two-finger grippers lack the dexterity for multi-page grasping, which requires counting layers before grasping them simultaneously.

To tackle these challenges, we propose *RoTipBot*, a robotic system using rotatable tactile sensors to detect contact, feed multiple layers to the center, and facilitate grasping and counting. We introduce *RoTip* [7], a vision-based tactile sensor that senses the entire fingertip area and actively rotates. As shown in Fig.1-(a), its sensing capability reduces visual noise and ensures reliable contact, while its rotational motion adds dexterity. A segmentation-projection model extracts contact areas and surface planes for grasp planning. As shown in Fig.1-(b), RoTipBot's passive finger holds the object while the active finger rotates to gather layers (Both fingers support

Jiaqi Jiang is with the School of Aerospace Engineering, Beijing Institute of Technology, Beijing 100081, China, and was with the Department of Engineering, King's College London, WC2R 2LS, U.K.

Xuyang Zhang, Daniel Fernandes Gomes and Shan Luo are with the Department of Engineering, King's College London, WC2R 2LS, U.K.

Thanh-Toan Do is with the Department of Data Science and AI, Monash University, Clayton, VIC 3800, Australia.

∗represents equal contributions.

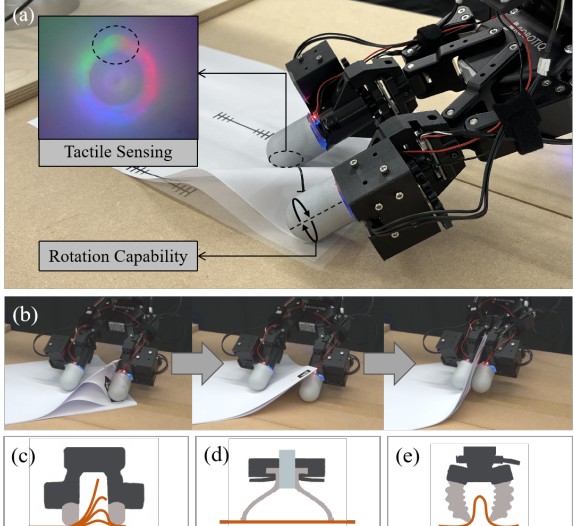

Fig. 1. **(a)** A demonstration of RoTipBot. The tactile sensors ensure good contact with objects, while the rotation capability feeds multiple layers of thin, flexible objects into the centre for grasping and counting. Different transparencies of the paper represent states at different time steps. **(b)** Snapshots of the feeding process for multiple print papers. **(c-e)** Sketches comparing RoTipBot to approaches based on suction cups and soft grippers. RoTipBot can count multiple layers and then grasp them simultaneously in a single grasp closure, whereas the other methods cannot.

indefinite rotation at 5 rad/s, max 12.9 rad/s). The fingers continuously adjust to stack thickness, maintaining contact and ensuring accurate multi-layer grasping. Tactile sensing further enables precise paper counting. This work has been accepted and published in IEEE Transactions on Robotics [8], where a more detailed discussion of related tactile roller grippers [9], [10] can be found.

## II. ROTIPBOT SYSTEM

RoTipBot uses a two-finger gripper to handle thin and flexible objects: one **passive** finger holds the object, while the other **active** finger rotates to feed multiple sheets into the centre between the two fingers, allowing them to be grasped all at once.

RoTipBot follows a structured process for handling thin and flexible objects, as shown in Fig. 2. First, during the vision-based grasping generation stage, an RGB-D image captured by a camera is used to generate grasp proposals, guiding the robot to make contact and grasp the object. Next, in the tactile-based adjustment for two-finger sufficient contact stage, RoTip's sensing capabilities compensate for visual perception noise by refining the end-effector's position. Once both RoTip

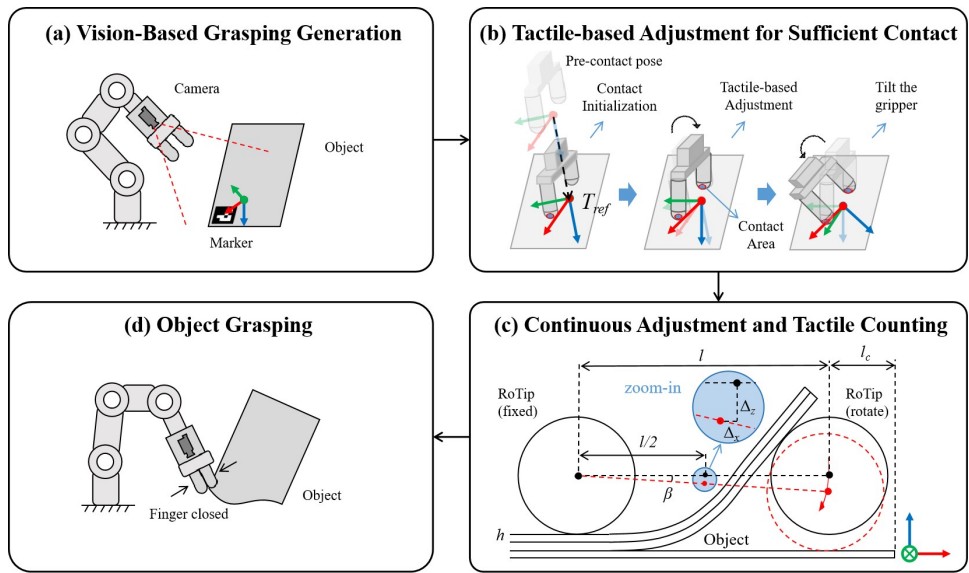

Fig. 2. An overview of our RoTipBot for thin and flexible handling. (a) Vision-Based Grasping Generation. (b) Tactile-based Adjustment for Two-finger Sufficient Contact. (c) Continuous Adjustment and Tactile Counting. (d) Object Grasping.

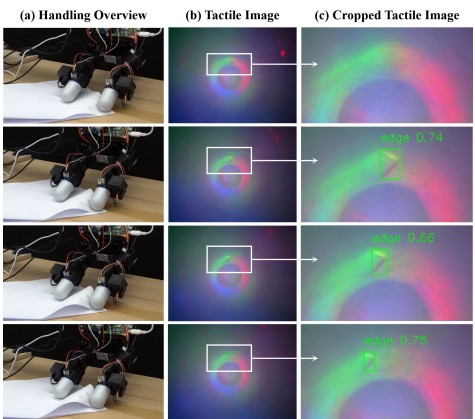

Fig. 3. Demonstration of page counting with tactile feedback. (a) The vision snapshots during feeding the stacked papers; (b) The paper edges can be captured in the tactile images to count the number of fed papers. (c) The zoomed-in contact areas in tactile images.

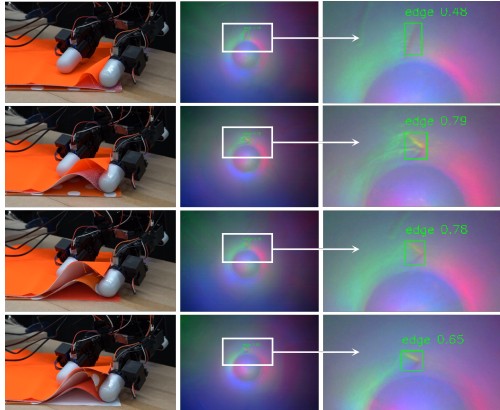

Fig. 4. Snapshots of grasping multi-layer fabrics. RoTipBot is capable of grasping multiple layers of fibrics composed of different materials with a nearly 100% success rate, while simultaneously counting the number of layers grasped.

sensors make contact with the object, the end-effector rotates around its $x$-axis to achieve an optimal inclination for feeding and grasping. Then, in the continuous adjustment and tactile counting stage, a continuous pose adjustment strategy ensures stable two-finger contact while feeding multiple thin, flexible layers. Simultaneously, tactile sensing is employed to count the number of layers being fed, a process known as tactile counting. Finally, in the object grasping stage, the gripper closes to securely pick up the objects.

Tactile sensing is also employed to count the number of fed pages, a process referred to as *tactile counting*, as shown in Fig. 4. A YOLOv11 model is used to detect the paper boundaries including edges and corners during the feeding process of multiple thin and flexible sheets. The detection focuses on the edges of the paper to track movement. When the centre of a detected edge crosses a predefined threshold, the paper is marked as "held" for rolling by the RoTipBot. The count increases by 1 when a new edge appears, indicating a new paper is being rolled by the sensor.

Tactile-based object counting could be highly beneficial for tasks such as flipping to a specific page range in a book by detecting the number of pages through touch. This method relies on physical feedback, making it less dependent on lighting conditions, which provides a significant advantage in environments where book indices are obscured or unavailable. RoTipBot achieved grasp success rates of 96%, 93%, and 97%, and counting success rates of 97%, 94%, and 98% on printer paper, coated paper, and plastic pocket sheets, respectively, detailed in [8].

## III. Conclusion

In this paper, we propose RoTipBot, a novel approach for handling thin and flexible objects using rotatable tactile sensors. It is the first approach that can count multiple layers and then grasp them simultaneously in a single grasp closure. The limitations of this work include difficulties in grasping highly wrinkled sheets and objects with high interlayer friction. Future work includes developing general manipulation strategies through reinforcement learning and integrating advanced vision algorithms to enhance applicability.

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
