# OpenReview forum: "RoTipBot: Robotic Handling of Thin and Flexible Objects using Rotatable Tactile Sensors"
_IEEE.org/IROS/2025/Workshop/Tactile_Sensing — IROS 2025 Workshop Tactile Sensing OralPoster_

### Official Review · Reviewer_vzJd · 2025-09-13
**Very interesting work, a few suggestions for completeness**

**Rating:** 9
**Confidence:** 5

**Review:**

This is a very interesting work. The authors present RoTipBot, a two-finger gripper designed to manipulate thin, flexible objects. Its key design combines an active, rotatable finger with vision-based tactile feedback and a passive finger, enabling the system to grasp and count multiple layers of thin objects simultaneously in a single grasp closure.
To make this extended abstract more self-contained, it would be helpful to summarize the system’s performance metrics (e.g., success rate, counting accuracy), even though they are detailed in the original T-RO paper. A brief mention of the limitations and future plans would be a great addition, too.

---

### Official Review · Reviewer_LWUz · 2025-09-15
**Review on RoTipBot**

**Rating:** 10
**Confidence:** 4

**Review:**

The paper introduces an elegant and original tactile manipulation system based on a fingertip-shaped tactile sensor capable of rotation. The combination of hardware design and control algorithms enables impressive capabilities such as page counting. Overall, the work is inspiring, well-written, and makes a valuable contribution to tactile manipulation research. I have a few suggestions and clarifications:

1) Connection to related work on tactile roller grippers

It would be useful to briefly mention the relation to tactile roller grippers, for example:

Pan, Chaoyi, et al. "In-hand manipulation of unknown objects with tactile sensing for insertion." 2023 IEEE/RSJ International Conference on Intelligent Robots and Systems (IROS). IEEE, 2023.

This would help position RoTipBot in the context of other rolling/rotating tactile manipulation approaches.

2) Clarification on sensor actuation

It is not entirely clear whether both fingertip sensors are capable of rotation, and if so, whether only one is actuated at a time by design. Could the system also benefit from coordinated actuation where both rotate simultaneously, and if so, what types of tasks would that enable?

3) Rotation capabilities of the tactile sensor

It would be helpful to specify whether the tactile sensor can rotate indefinitely or if there is a mechanical joint/rotation limit.

Additionally, reporting the average and maximum fingertip rotation velocities would give readers a better sense of the manipulation dynamics and speed.